# Practice Patterns in Fournier’s Gangrene in Europe and Implications for a Prospective Registry Study

**DOI:** 10.3390/antibiotics12020197

**Published:** 2023-01-18

**Authors:** Laila Schneidewind, Bernhard Kiss, Fabian P. Stangl, Zafer Tandogdu, Florian M. E. Wagenlehner, Truls E. Bjerklund Johansen, Béla Köves, Jose Medina-Polo, Ana Maria Tapia, Jennifer Kranz

**Affiliations:** 1Department of Urology, University Medical Center Rostock, 18057 Rostock, Germany; 2Department of Urology, University Hospital of Bern, 3010 Bern, Switzerland; 3Department of Urology, University College London Hospitals, London NW1 2BU, UK; 4Clinic for Urology, Pediatric Urology and Andrology, Justus-Liebig-University Giessen, 35390 Giessen, Germany; 5Department of Urology, Oslo University Hospital, Institute of Clinical Medicine, University of Oslo, 0315 Oslo, Norway; 6Department of Urology, University of Szeged, 6720 Szeged, Hungary; 7Department of Urology, Hospital Universitario 12 de Octubre, 28041 Madrid, Spain; 8Department of Urology, Hospital Universitario Río Hortega, 47012 Valladolid, Spain; 9Department of Urology and Pediatric Urology, University Medical Center RWTH Aachen, 52074 Aachen, Germany; 10Department of Urology and Kidney Transplantation, Martin Luther University, 06108 Halle (Saale), Germany

**Keywords:** Fournier’s gangrene, sepsis, necrotizing fasciitis, outcome parameters, antibiotic stewardship

## Abstract

Background: Fournier’s gangrene (FG) is a life-threatening, necrotizing infection. Due to the rareness of the disease, it is challenging to plan robust prospective studies. This study aims to describe current practice patterns of FG in Europe and identify implications for planning a prospective FG registry. Methods: Online non-validated 17-items survey among urologists treating FG in in European hospitals. Questionnaires were analyzed with LimeSurvey (LimeSurvey GmbH Hamburg, Germany). Results: 229 responses from ten different European countries were submitted, and 117 (51.1%) urologists completed the questionnaire. The departments treat a mean of 4.2 (SD 3.11) patients per year. The urology department mostly takes the lead in treating FG patients (*n* = 113; 96.6%). The practice in FG is very heterogenic and mostly case-based all over Europe, e.g., vacuum-assisted wound closure (VAC) is mostly used (*n* = 50; 42.7%) as adjunct wound. The biggest challenges in FG are the short time to diagnosis and treatment, standardization and establishment of guidelines, and disease awareness. Additionally, participants stated that an international registry is an outstanding initiative, and predictive models are needed. Conclusions: There is no standard of care in the diagnosis, treatment, and long-term care of FG all over Europe. Further research could be conducted with a prospective registry.

## 1. Introduction

Fournier’s gangrene (FG) is a sporadic, life-threatening, necrotizing, bacterial infection affecting the perineum, perineal region, and genitals [1,2,3]. Hence, the incidence is very low, and most of the limited knowledge about FG arises from retrospective single-institutional studies with very small patient cohorts [4,5,6,7,8]. The incidence of FG is 1.6 cases per 100,000 male patients in the United States [9]. FG affects all ages and sexes, with a strong male preponderance [6]. The most important reported comorbidity in FG is diabetes mellitus [10]. Furthermore, the prognosis, survival, and outcome of FG have not improved in recent years, despite more intensive critical-care therapy for those patients [3,11]. In their multicenter retrospective study of 154 cases, Kranz et al. showed that survival time did not improve in recent years (*p* = 0.268), with up to 15.4% of the patients dying during inpatient treatment [3]. Development of FG is often associated with an infection nidus in the genitourinary tract. The disease often progresses in a rapid fashion, subsequently causing an immense inflammatory response. This leads to multiple organ dysfunction, septic shock, and death if left untreated. Bacterial isolates show a polymicrobial flora in over 80% of cases with a multitude of aerobic and anaerobic bacteria [12]. The most common organisms isolated are *Escherichia coli*, *Klebsiella pneumonia*, *Staphylococcus aureus,* and *Bacteroides fragilis*. Diagnosis of FG is primarily made by identifying clinical findings such as the characteristic crepitus and extremely tender lesions. In the early phase of FG, pain may seem out of proportion compared to the clinical findings [13]. After identifying FG, clinicians should not delay treatment which consists of hemodynamic resuscitation, broad spectrum antibiotics after sampling and aggressive, wide surgical debridement [14].

Fournier’s gangrene (FG) is a sporadic, life-threatening, necrotizing, bacterial infection affecting the perineum, perineal region, and genitals [1,2,3]. Hence, the incidence is very low, and most of the limited knowledge about FG arises from retrospective single-institutional studies with very small patient cohorts [4,5,6,7,8]. The incidence of FG is 1.6 cases per 100,000 male patients in the United States [9]. FG affects all ages and sexes, with a strong male preponderance [6]. The most important reported comorbidity in FG is diabetes mellitus [10]. Furthermore, the prognosis, survival, and outcome of FG have not improved in recent years, despite more intensive critical-care therapy for those patients [3,11]. In their multicenter retrospective study of 154 cases, Kranz et al. showed that survival time did not improve in recent years (*p* = 0.268), with up to 15.4% of the patients dying during inpatient treatment [3]. Development of FG is often associated with an infection nidus in the genitourinary tract. The disease often progresses in a rapid fashion, subsequently causing an immense inflammatory response. This leads to multiple organ dysfunction, septic shock, and death if left untreated. Bacterial isolates show a polymicrobial flora in over 80% of cases with a multitude of aerobic and anaerobic bacteria [12]. The most common organisms isolated are *Escherichia coli*, *Klebsiella pneumonia*, *Staphylococcus aureus,* and *Bacteroides fragilis*. Diagnosis of FG is primarily made by identifying clinical findings such as the characteristic crepitus and extremely tender lesions. In the early phase of FG, pain may seem out of proportion compared to the clinical findings [13]. After identifying FG, clinicians should not delay treatment which consists of hemodynamic resuscitation, broad spectrum antibiotics after sampling and aggressive, wide surgical debridement [14].

Summarizing, key points for the successful treatment of FG are immediate surgical debridement, accompanied by intensified antibiotic therapy, and intensive care medical management [15]. However, further research to improve the outcome of FG is desperately needed [16]. Despite all above-mentioned measures and advances, mortality remains high.

Unfortunately, due to the rareness of the disease, it is challenging to perform robust prospective clinical studies. In our working group, we have already completed a survey about the practice patterns in diagnostics and treatment of FG in German academic medicine to describe the situation and identify implications for planning a prospective clinical registry. Overall, the authors concluded that the contemporary practice patterns in FG are very heterogenous, but the outcome is still problematic, and the disease is difficult to predict. Additionally, some essential points for a registry study, such as histological confirmation of the disease have been identified [16].

Consequently, a survey about practice patterns in FG in European hospitals to generate more valid data for a prospective European registry study for FG was conducted, which might improve outcome of this severe disease. This study’s primary aim was to describe current practice patterns in European hospitals. The secondary aim was to identify factors associated with a higher mortality rate, and we attempted to use the data for the detailed planning of the registry study.

## 2. Results

### 2.1. Baseline Characterization

About 229 responses from ten different European countries were submitted, and 117 (51.1%) totally completed the questionnaire. Figure 1 presents an overview about the participation of the ten different countries on complete responses, with Germany (*n* = 40; 34.2%), Spain (*n* = 20; 17.1%), and Austria (*n* = 14; 11.9%) being on top. The departments treat a mean of 4.2 (SD 3.11) patients per year. Mostly, the urology department takes the lead in the treatment of FG patients (*n* = 113; 96.6%), followed by general surgery (*n* = 2; 1.7%) and gynecology (*n* = 2; 1.7%).

### 2.2. Immediate Management

In most cases, there is no standard operating procedure (SOP) for the therapy of FG (*n* = 85; 72.6%), while 32 departments (27.4%) have one. Furthermore, this SOP is a departmental one (*n* = 22; 68.8%), a local area one (*n* = 6; 18.7%), or a national one (*n* = 4; 12.5%). Urology is often on the lead in the debridement team (*n* = 85; 72.6%) or the debridement surgical team is case-based (*n* = 32; 27.4%). There is an initial empiric standard antibiotic therapy in 68 of the cases (58.1%). Most often a combination of metronidazole plus cephalosporine plus aminoglycoside was used (*n* = 26; 38.2%), followed by piperacillin/tazobactam (*n* = 22; 32.4%) and carbapenems (*n* = 20; 29.4%).The departments added metronidazole to piperacillin/tazobactam or carbapenem in four cases. A transurethral catheter (*n* = 51; 43.6%) is the most frequent urinary diversion, twenty-one (17.9%) use a suprapubic catheter and thirty-three (28.2%) use both routes of catheterization. Twelve (10.3%) participants stated that they use a different urinary diversion, which is often case-based. Vacuum-assisted wound closure (VAC) is mostly used (*n* = 50; 42.7%) as adjunct wound therapy, 41 (35.0%) do not use an additional therapy for the wound, and in 17 (14.5%) of the cases, hyperbaric oxygenation (HBO) is used. Nine (7.8%) participants reported using something different as adjunct wound therapy, which is case based. This adjunct wound therapy is mostly dedicated to selected cases (*n* = 49; 64.5%). Twenty-four comments on that issue were received: Twenty-two participants (91.7%) stated that adjunct wound therapy depends on the local wound situation and size; two (8.3%) that wound therapy relies on patient compliance. In summary, the consensus of the comments was: the bigger the wound of FG is, it is more likely to receive an adjunct wound therapy, and then VAC is mostly used. Interestingly, FG severity index (FGSI) is not used in daily clinical practice (*n* = 107; 91.5%).

### 2.3. Reconstructive Approach

In most urological departments, the reconstruction is done by a plastic surgeon (*n* = 85; 72.6%). Furthermore, it was asked if the plastic surgeon is always needed or only in particular cases, so the plastic surgeon is considered in specific cases (*n* = 55; 64.7%) only, and this selection is depending on the case and wound situation.

### 2.4. Prognostic and Survival Factors

Fifteen survey participants (12.8%) observed an association of FG with SGLT2 inhibitors or other anti-diabetic medication. Additionally, thirteen (11.1%) commented that: five (4.3%) observed an association with oral anti-diabetic drugs, three (2.6%) with metformin, two (1.7%) with SGLT2 inhibitors, one (0.8%) with dapagliflozin (also a SGLT2 inhibitor), and one (0.8%) with insulin.

Furthermore, it was asked for 30- and 90-day mortality. The received data were inconclusive and only estimated.

### 2.5. Open Comments and Implications for Improvement

One open question was asked: what is the biggest challenge in FG which could be improved? Eighty-two (70.1%) of the survey participants commented on that. Some of the participants even had several ideas and suggestions. Table 1 gives an overview of the most critical challenges and how often they were mentioned.

In summary, most colleagues agree that there must be a brief period from the emergency room to diagnosis and treatment of FG to achieve successful outcomes. This can only be accomplished by strengthening the awareness and education about FG in urology departments and among nurses, residents, colleagues from other specialties and medical co-workers. In that respect, one participant had an excellent idea to establish training videos about FG and correct debridement for residents, nurses, other medical professionals, and even patients. However, there is also an urgent call for standardization and establishing guidelines for FG.

In the open remarks for this survey, we received many comments for improvement of our questionnaire and planning an international registry in our online tool (*n* = 31; 25.6%). Additionally, participants stated that an international registry is an outstanding initiative to gain more insights on this infection. Two participants (1.7%) stated that this should also focus on biomarkers for survival prediction and two (1.7%) suggested that further studies should also address sepsis and intensive care management.

## 3. Discussion

We conducted a European Survey about the therapy situation of FG in treating hospitals, mainly to generate data to plan a robust FG registry study for improving outcomes. To our knowledge, this is the first survey of this kind in European hospitals. Luckily, physicians from ten different countries participated, generating an excellent first overview of FG in Europe. Furthermore, the results for baseline/demographic characterization are not surprising at all. FG is still a rare disease, with four cases per year in most hospitals, and most urologists take the lead in treating this disease. As mentioned above, it is challenging to plan prospective clinical trials in rare diseases and it is known that in these diseases research is insufficient [17]. This calls urologists as the leading treating physicians into action to conduct proper research, e.g., with a prospective registry [16].

Concerning the immediate management of FG, the situation all over Europe is very heterogeneous and often case-based, which is also unsurprising, but there are two critical issues to discuss. First, the significance of adjunct wound therapies such as VAC and HBO. There was a consensus in our survey: The bigger the wound of FG is, the more likely it is to receive an adjunct wound therapy, and then VAC is mainly used. However, it is unclear if adjunct treatment can improve outcomes and when or in which patient groups it is functional. In our opinion, this can be answered with a prospective registry, too, especially model systems that could be developed to predict disease better [16]. Second, it is well known that FG progression is very hard to predict [18,19]. One prediction tool is the FGSI, based on the parameters temperature, heart rate, respiratory rate, serum sodium, serum potassium, serum creatinine, hematocrit, white blood cell count, and serum bicarbonate [20]. Unfortunately, it is hardly used in daily clinical practice all over Europe. Thereof, developing a more robust and valid prediction tool for FG outcome for clinical practice would be of value. Prediction models could be easily integrated into an FG registry online platform, where it also can be further improved with every patient, which is added.

These considerations also lead us to the reconstructive approach, which is also heterogenic and case-based all over Europe. Which patient will benefit from special wound treatments and reconstruction performed by a plastic surgeon? A predictive model system might also be helpful in this case. Additionally, concerning the wound situation outcome means also long-term results and quality of life. Quality of life data about FG are also sparse [18,19], but they are vital for the patients [21]. Consequently, quality of life data should also be gathered into a registry study and considered important factors for long-term clinical outcomes.

Unfortunately, our data about prognostic and survival factors are misleading, but this leads to the obvious conclusion that FG is very difficult to predict [20]. Furthermore, we must discuss that the mentioned association with diabetic medication and SGLT2 inhibitors is biased. There is one systematic review with meta-analysis including 84 studies, which concluded that there is no significant association of FG with SHLT2 inhibitors [22], so the physician in daily clinical practice might see many patients with diabetic medication and severe infection, but that might be only a confounder and the diabetes mellitus is just not treated heavily enough, which can lead to severe infections.

In our opinion, the most important results of this survey arise from the open comments and implications for improvement. On the whole, a prospective international FG registry study is mentioned as an outstanding initiative, which can also integrate biomarkers for survival prediction, sepsis, and intensive care management. The three most significant challenges in FG are short time to diagnosis and treatment, standardization and establishment of guidelines, and awareness of the disease. The time to treatment is the first critical point for FG outcome. Baser et al. showed in their retrospective study of 66 patients that the waiting period in the emergency room holds a diagnostic value in predicting FG mortality (*p* < 0.0001). Their critical cut off waiting period was 136 min [23]. Consequently, how can clinicians shorten the time to diagnose and treat FG? The first step is to be aware of this rare disease, which was also a major challenge in our survey. Physicians must educate all our residents, colleagues, nurses, and medical co-workers in FG. This education tools could also be integrated into an online platform of the registry, e.g., learning videos, photo material, or summaries of evidence. Interestingly, this was also suggested by a participant. Overall, education tools are also important to tackle FG. Lastly, standardization and establishment of guidelines can only be achieved with an improvement of research and the generation of robust data. As mentioned several times, this is difficult in rare diseases such as FG because of the low number of cases for prospective clinical studies. In summary, a prospective registry study is very reasonable and can be amended by learning online predictive tools using data integration and educational tools, even shortcuts for the emergency room. This survey provides important new aspects and implications for the development of such a registry.

Self-explanatory, this study has several limitations, such as the non-validated questionnaire and the small sample size concerning whole Europe, so there is a risk of selection bias.

However, the key point to discuss is the essential research need in FG, especially regarding the improvement of outcome and the development of more practical prognostic systems. First step here is the improvement of outcome. In our opinion, the identification of patients who will benefit from special adjunct therapies, such as VAC or HBO, is absolutely warranted. Unfortunately, it is well-known that research and improvement of treatment of rare diseases is a problem in health care. One approach to solve that problem is the European Reference Network eUROGEN for rare urogenital diseases and complex conditions in both children and adults [24]. For this reason, an online platform with registry study is the right approach to tackle FG and improve outcome for our patients.

## 4. Materials and Methods

### 4.1. Development of the Survey and Target Population

The survey was conducted among specialists in infections in urology. It was designed and conducted according to the reporting guidelines for surveys found on the equator-network.org, an international initiative providing robust reporting guidelines [25,26]. The first step for the development of the survey was item generation. An explorative literature search on MEDLINE via PubMed was done using the MeSH term “Fournier´s gangrene” in order to identify key topics and questions concerning FG. Furthermore, the authors used our experience, results and already developed a non-validated questionnaire from the German survey on FG in University Medical Centers [16]. In summary, a 17-item questionnaire with 15 items being multiple choice (in three cases we asked for specification if the answer was yes) and two open questions was developed. The questions included information about baseline characteristics, immediate management of FG, prognostic factors, and remarks for improvement. The target population were urologists treating FG in European hospitals. The aim was to only get one response from one hospital.

### 4.2. Administration of the Survey

The questionnaire was transferred to the online platform LimeSurvey (LimeSurvey GmbH, Hamburg, Germany) and tested for usability by the co-authors. Afterwards it was set online on 1 February 2022. An invitation was sent out via ESIU (European Section of infections in urology) and all European contacts for the ESIU members as well as the co-authors. Overall, six reminders were sent out in three months. The survey was set offline on 30 April 2022. All procedures performed in this study were in accordance with the ethical standards of the institutional and/or national research committee and with the 1964 Helsinki Declaration and its later amendments or comparable ethical standards. For this type of study, a formal consent was not required.

### 4.3. Statistical Analysis

Only complete questionnaires were analyzed with LimeSurvey, except the last open question asking for remarks and comments for improvement. Here, all available information was used. For each numeric variable, the numeric distribution was preliminarily assessed by the Kolmogorov–Smirnov test. Descriptive statistics were made with mean and standard deviation (SD) for normal distribution or with median and interquartile range (IQR) for non-parametric data.

## 5. Conclusions

In summary, FG is a rare but very severe disease. The outcome has not improved significantly in recent years. There is no standard of care in the diagnosis, treatment, and long-term care of FG all over Europe. Additionally, the disease is very difficult to predict. The current biggest challenges are awareness of FG and time to treatment and the standardization of care. Consequently, further robust research on a rare disease is essential. This can be done with a prospective international online registry. Due to modern data integration solutions, this registry could be accompanied by online predictive tools and educational information.

## Figures and Tables

**Figure 1 antibiotics-12-00197-f001:**
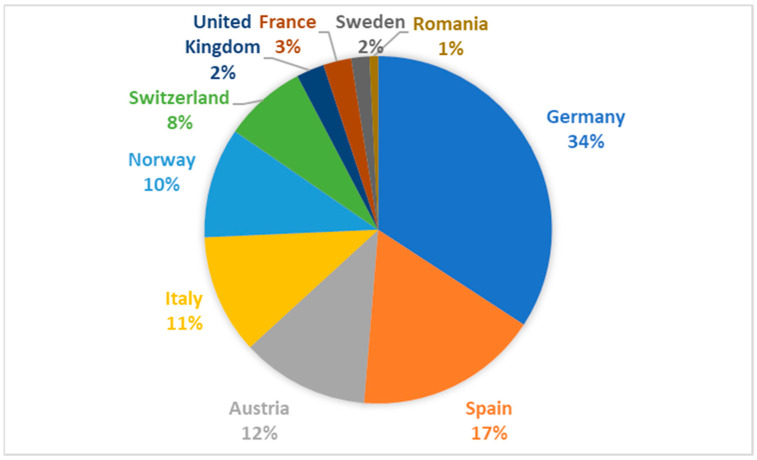
Percental participation from ten different European countries.

**Table 1 antibiotics-12-00197-t001:** Biggest challenges in FG and their importance to the survey participants.

Challenge	Number of Mentions
Short time to diagnosis and treatment	44
Standardization and establishment of guidelines	22
Awareness of the disease	20
Strengthen the awareness of FG in the different specialties	9
Reconstructive Surgery	6
Postoperative wound care	5
Pre-treatment morbidity	3
Antibiotic Stewardship	3
Prophylaxis	2
Adequate debridement	1
Correct visual diagnosis	1
Collaboration between departments	1
Intensive care management	1
Recovery period	1

## Data Availability

All data can be assessed with formal request to the corresponding author.

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
