# Peer review of "Practice Patterns in Fournier’s Gangrene in Europe and Implications for a Prospective Registry Study"

_antibiotics, 2023, doi:10.3390/antibiotics12020197_

Round 1
Reviewer 1 Report
Thank you for reviewing this manuscript. The authors discuss the rare Fournier’s gangrene, a potentially life-threatening disease. They made an online questionnaire and involved ten urological centers from Europe. They investigated essential questions, e.g. immediate disease management, anatomical reconstruction approach, survival, and prognosis. This is a very important study; I have only one question for the authors.
It should be clarified why only ten urology centers were involved in this online questionnaire or what was the selection criteria?
With the number of participants, this disease seems more frequent in Germany. Is this true?
Author Response
Thank you very much for your kind review of our article. The centers chosen to participate in our study is due to the association of the members of our research work group. In order to facilitate response in a timely manner with physicians heavily involved in infectious diseases, we did roll out our study further.
The participation was diverse between countries which may be due to advertisement of the questionnaire. Local meetings in Germany allowed for better direct advertisement.
Reviewer 2 Report
1. In introduction, the reference cited pattern should be same. Check line number 45 and 47.
2. In introduction, there should be some details about the disease, their risk factors, possible treatment options, their pattern in Europe. Introduction should be aligned with connective sentences.
3. Figure 1 is unclear; suggestion is to change the chart style in which percentage mentioned.
3. Table 1 is not related with the study. Results must be in tabular form rather than text or descriptive. The aims and objective are something different than results. No statistical analysis in study?
4. what will be specialty of urologist?
5. How you validated the questions?
6. Any compensations you provided to the doctors for filling the form?
7. Overall, English sentences have incomplete sense.
Author Response
Thank you very much for your extensive review. We tried to address all the issues thoroughly point by point.
- We edited the citation
- We added the requested details
- We edited the figure accordingly
- We are not certain what urologist you are referring to, but urologist involved in infectious diseases participated
- The questionnaire was designed according to mentioned guidelines. The creators of the questionnaire firstly conducted two trial-runs among a smaller group of participants to review validity of our questionnaire.
- No compensation was provided
- We are uncertain to what sentences you are referring to. The manuscript was written and corrected by native speakers (UK)
Reviewer 3 Report
Schneidewind et al., reported an interesting work in the research article entitled "Practice patterns in Fournier´s gangrene in Europe and implications of a prospective registry study". The current study will be very important in the medical field. No doubt the manuscript is very informative and relevant. I have made several suggestions for the authors:
1. Try to avoid using, I, or We in the manuscript.
2. The quality of Fig.1 is very poor. It must be improved.
3. Table 1 must be improved. It is cut into two pages.
4. Current manuscript is too short for considering it as an article. It must be a brief report or must be elaborated.
Conclusion: Major revision.
Author Response
Thank you very much for your comments. We addressed all your concerns.
To 1: We amended the manuscript accordingly.
To 2: We improved the figure..
To 3: We are uncertain regarding your comment. In the submitted word document the table only spans one page. But, if necessary, we will downsize the table.
To 4: According to the author guidelines original articles must be around 3000 words to be considered. Our current edited article surpasses this benchmark. We hope you deem this sufficient.
Reviewer 4 Report
The article ``Practice patterns in Fournier's gangrene in Europe and implications a prospective registry study'' is a well and correctly methodologically done study, with significant results, beautifully presented. In the discussion, the topic with the results is handled correctly. the conclusion is in line with the research.
Author Response
Thank you very much for your kind comment. We appreciate it!
Round 2
Reviewer 2 Report
Paper looks good after modification. It may be accepted.
Reviewer 3 Report
The manuscript can be accepted in the present form.